# The impact of chest CT body composition parameters on clinical outcomes in COVID-19 patients

Giulia Besutti[1,2], Massimo Pellegrini[3,4]*, Marta Ottone[5], Michele Cantini[6], Jovana Milic[2,6], Efrem Bonelli[1,7], Giovanni Dolci[8], Giulia Cassone[2,9], Guido Ligabue[10], Lucia Spaggiari[1], Pierpaolo Pattacini[1], Tommaso Fasano[7], Simone Canovi[7], Marco Massari[8], Carlo Salvarani[9], Giovanni Guaraldi[6], Paolo Giorgi Rossi[5], on behalf of the Reggio Emilia COVID-19 Working Group[¶]

1 Radiology Unit, Azienda USL–IRCCS di Reggio Emilia, Reggio Emilia, Italy, 2 Clinical and Experimental Medicine PhD Program, University of Modena and Reggio Emilia, Modena, Italy, 3 Clinical Nutrition Unit, Azienda USL–IRCCS di Reggio Emilia, Reggio Emilia, Italy, 4 Department of Biomedical, Metabolic and Neural Sciences, University of Modena and Reggio Emilia, Modena, Italy, 5 Epidemiology Unit, Azienda USL–IRCCS di Reggio Emilia, Reggio Emilia, Italy, 6 Modena HIV Metabolic Clinic, University of Modena and Reggio Emilia, Modena, Italy, 7 Department of Diagnostic Imaging and Laboratory Medicine, Clinical Chemistry and Endocrinology Laboratory, Azienda USL–IRCCS di Reggio Emilia, Reggio Emilia, Italy, 8 Infectious Disease Unit, Azienda USL–IRCCS di Reggio Emilia, Reggio Emilia, Italy, 9 Rheumatology Unit, Azienda USL–IRCCS di Reggio Emilia, Reggio Emilia, Italy, 10 Radiology Unit, Azienda Ospedaliero-Universitaria di Modena, University of Modena and Reggio Emilia, Modena, Italy

¶ Membership of the Reggio Emilia COVID-19 Working Group is provided in the Acknowledgments.
* massimo.pellegrini@unimore.it

**Data Availability Statement:** According to Italian law, anonymized data can only be made publicly available if there is potential for the re-identification

## Abstract

We assessed the impact of chest CT body composition parameters on outcomes and disease severity at hospital presentation of COVID-19 patients, focusing also on the possible mediation of body composition in the relationship between age and death in these patients. Chest CT scans performed at hospital presentation by consecutive COVID-19 patients (02/27/2020-03/13/2020) were retrospectively reviewed to obtain pectoralis muscle density and total, visceral, and intermuscular adipose tissue areas (TAT, VAT, IMAT) at the level of T7-T8 vertebrae. Primary outcomes were: hospitalization, mechanical ventilation (MV) and/or death, death alone. Secondary outcomes were: C-reactive protein (CRP), oxygen saturation (SO2), CT disease extension at hospital presentation. The mediation of body composition in the effect of age on death was explored. Of the 318 patients included in the study (median age 65.7 years, females 37.7%), 205 (64.5%) were hospitalized, 68 (21.4%) needed MV, and 58 (18.2%) died. Increased muscle density was a protective factor while increased TAT, VAT, and IMAT were risk factors for hospitalization and MV/death. All these parameters except TAT had borderline effects on death alone. All parameters were associated with SO2 and extension of lung parenchymal involvement at CT; VAT was associated with CRP. Approximately 3% of the effect of age on death was mediated by decreased muscle density. In conclusion, low muscle quality and ectopic fat accumulation were associated with COVID-19 outcomes, VAT was associated with baseline inflammation. Low muscle quality partly mediated the effect of age on mortality.

of individuals (https://www.garanteprivacy.it). Furthermore, property of the data remains of the patient, who gave consent to use data for the objective of the study. Thus, the data underlying this study are available on request to researchers who intend to conduct research in the respect of confidentiality (even if anonymous data are provided, they should be published in aggregated form) and for studies with objectives consistent with those of the original study. In order to obtain data, approval must be obtained from the Area Vasta Emilia Nord (AVEN) Ethics Committee, who would check the consistency of the objective and planned analyses and would then authorize us to provide aggregated or anonymized data. Data access requests should be addressed to the Ethics Committee at CEReggioemilia@ausl.re.it as well as to the authors at the Epidemiology unit of AUSL - IRCCS of Reggio Emilia at info.epi@ausl.re.it, who are the data guardians.

**Funding:** This study is part of a larger project supported by Ministry of Health (Grant number COVID-2020-12371808).

**Competing interests:** The authors have declared that no competing interests exist.

# Introduction

A novel severe acute respiratory syndrome Coronavirus 2 (SARS-CoV-2) has recently emerged as a global health threat [1, 2]. As of 15 February, over 108 million people had been affected with more than 2.390.000 deaths reported worldwide so far [3]. The case fatality rate varies dramatically across countries and phases of the epidemic, ranging from 2% to 20%, depending on the characteristics of the population and the ability of the health system to identify less severe cases. Most severe COVID-19 patients develop acute respiratory distress syndrome (ARDS) or sepsis with multiorgan dysfunction [1, 4, 5], often associated with an uncontrolled cytokine-mediated immune response called the *cytokine storm*. Of these patients, 71–75% need assisted mechanical ventilation and about 50% die [1, 2, 5–7]. Obesity and advanced age are among the most important recognized risk factors for an unfavorable outcome in COVID-19 patients [1, 2, 8–10].

Obesity was considered a risk factor during the previous H1N1 virus outbreak as well [11], and it is not surprising that SARS-CoV-2 pneumonia is also negatively affected by overweight [12]. A higher percentage of body fat mass is associated with a greater cardiometabolic risk, but not all body fat deposits have the same significance. While body mass index (BMI) represents a useful but rough index of general adiposity, ectopic visceral, hepatic, and muscular fat depots are associated with increased production of pro-inflammatory cytokines and with a higher incidence of cardiovascular events, insulin resistance, and type 2 diabetes [13, 14]. In this *milieu*, the onset of the COVID-19 cytokine storm may be favored.

Advanced age, another important risk factor for the more severe forms of COVID-19, is characterized by a progressive reduction of body muscle mass and function (sarcopenia) associated with a progressive accumulation of fat deposits in the muscle (myosteatosis) [15, 16]. Both sarcopenia and myosteatosis impact medical and surgical outcomes and are reliable predictors of all-cause mortality [15, 17]. Body composition parameters are commonly studied by collecting CT cross-sectional areas at the level of L3 vertebra, linearly related to whole body muscle and fat mass [18]. Myosteatosis can be apparent within muscle fibers and is assessed through CT scans by measuring skeletal muscle density (SMD). In addition, it can be detected across muscle fibers and within the fascia, where it is assessed through CT scans by measuring the adipose tissue between muscles (intermuscular adipose tissue).

Recent publications have reported that higher BMI, higher abdominal visceral adipose tissue, higher intermuscular adipose tissue, reduced liver density, reduced lumbar SMD, and reduced pectoral muscle area are associated with worse clinical outcomes in COVID-19 patients in terms of disease severity and death [5, 19–25]. While CT scans of the abdomen are rarely available for unselected series of COVID-19 patients, chest CT has been widely used in some centers to rapidly assess the presence of pneumonia and stratify patients with different disease severity [26]. Body composition parameters measured on chest CT scans show moderate to high correlation both with abdominal CT fat compartments and skeletal muscle mass measured with bioelectrical impedance analysis [27, 28]. Moreover, like abdominal VAT, intrathoracic VAT (epicardial and extracardiac) is associated with the production of systemic inflammatory markers [29], and thoracic and pectoral muscle quantity and quality predict clinical outcomes in different respiratory diseases, especially in patients requiring mechanical ventilation [30–32].

The possible role of body composition parameters as prognostic factors for COVID-19 severity has been initially explored [5, 19, 20, 22–25]. However, the underlying causal relationship remains to be determined and contextualized in the complex pathogenetic pathways involved in COVID-19 progression. Therefore, we first investigated the association of chest CT-derived body composition parameters with clinical outcomes in COVID-19 patients,

including hospitalization, mechanical ventilation (MV) or death, and death alone. We also explored the association between body composition parameters and biomarkers of disease progression at emergency room presentation: oxygen saturation and extension of parenchymal involvement at CT for the lung damage, and C-reactive protein for the inflammatory reaction. Through a mediation analysis of the factors associated with age and death, secondly, we evaluated whether the effect of age on death is partly mediated by body composition.

## Materials and methods

### Setting

In the Reggio Emilia province (Northern Italy, 532,000 inhabitants, six hospitals), the first case of SARS-CoV-2 infection was diagnosed on 27 February 2020. As of 13 March 2020, there were 1,154 RT-PCR-confirmed COVID-19 patients in the province, with the daily number of new cases rising steadily.

### Study design and population

This observational study was approved by the Area Vasta Emilia Nord Ethics Committee on 7 April 2020 (protocol number 2020/0045199) and performed in accordance with the ethical standards of the Declaration of Helsinki. Given the retrospective nature of the study, the Ethics Committee authorizes the use of a patient's data without his/ her written informed consent if all reasonable efforts have been made to contact that patient.

All consecutive patients were included who presented to the provincial emergency rooms (ERs) between 27 February and 13 March 2020 for suspected COVID-19: these underwent chest CT at ER presentation and tested positive on RT-PCR for SARS-CoV-2 within 10 days. During the COVID-19 outbreak, virtually all symptomatic patients with suspected COVID-19 pneumonia were referred to CT. Patients with CT scans not suitable for different post-processing evaluations were excluded from specific study analyses, e.g., CTs with a small field of view were not suitable for evaluation of subcutaneous adipose tissue, CTs of patients with thoracic lipomas were excluded from the evaluation of fat compartments, and CTs with artifacts due to pacemakers or other implants were not suitable for pectoral muscle segmentation.

### Outcomes

The main outcomes considered were death, hospitalization, and death or mechanical ventilation while being a COVID-19 patient. We included all outcomes occurring between ER presentation and before symptom remission and two negative RT-PCR tests or end of follow up, i.e., 21 April 2020.

### Data collection

Date of symptom onset, diagnosis, hospitalization, and death were retrieved from the COVID-19 Surveillance Registry, coordinated by the Italian National Institute of Health and implemented in each Local Health Authority [33]. Registry data were linked with the hospital radiology information system to search for CTs performed at or after the onset of COVID symptoms and with hospital discharge databases to collect information on comorbidities. The Charlson Index was calculated based on hospital admissions in the previous 10 years [34]. BMI was calculated whenever patient height and weight registered within six months preceding COVID-19 diagnosis were available from the hospital information systems. Diabetes was ascertained through linkage with the local Diabetes Registry [35]. The need for invasive or non-invasive MV during hospitalization was manually collected from medical records.

## Blood tests

At ER presentation the levels of C-reactive protein, lactate dehydrogenase (LDH), white blood cell, lymphocyte, neutrophil, and platelet counts were routinely collected. Oxygen saturation level was also recorded for patients who had an arterial blood gas analysis before being provided with oxygen support. The tests were carried out in the Hospital Clinical Laboratories with routine automated methods.

## CT acquisition technique

CT scans were performed using one of three scanners (128-slice Somatom Definition Edge, Siemens Healthineers; 64-slice Ingenuity, Philips Healthcare; 16-slice GE Brightspeed, GE Healthcare) without contrast media injection, with the patient in supine position during end-inspiration. Scanning parameters were tube voltage 120 KV, automatic tube current modulation, collimation width 0.625 or 1.25 mm, acquisition slice thickness 2.5 mm, and interval 1.25 mm. Images were reconstructed with a high-resolution algorithm at slice thickness 1.0/1.25 mm.

## CT retrospective analysis

To evaluate COVID-19 pneumonia extension, CT scans were retrospectively reviewed by a chest radiologist with 15-year experience (LS), who graded extension of pulmonary lesions using a visual scoring system ($< 20\%$, $20–39\%$, $40–59\%$, $\geq 60\%$) [26].

To evaluate body composition parameters, CT images were retrospectively analyzed by a single trained image analyzer (EB) supervised by a senior radiologist (PP), both blinded to clinical data and outcomes, by using the OSIRIX-Lite software V5.0 (Pixmeo, Sarl, Switzerland) (S1 Fig).

As measures of sarcopenia, pectoralis muscle cross-sectional area ($cm^2$) and mean density (Hounsfield Unit, HU) were obtained selecting a single axial slice directly superior to the aortic arch and manually contouring both pectoralis major and minor on the right side (or on the left side when a defibrillator was present on the right), after applying a density range of -29 to 150 HU [36].

For total, subcutaneous, visceral, and intermuscular adipose tissue areas (TAT, SAT, VAT, and IMAT), a single slice at the level of the seventh to eighth thoracic vertebrae (T7-T8) was selected and a density range from -190 to -30 HU was applied. Fat compartments were measured through autosegmentation, with manual contour correction when necessary [27].

Mean liver and spleen attenuation values (HU) were obtained by drawing nine regions of interest (ROIs) in the liver and three ROIs in the spleen, paying attention to avoid vessels, bile ducts, focal lesions, focal fatty changes, and visceral margins.

For all retrospective measures, a second measurement was obtained in a sample of 15 consecutive patients by the same reader after two months, in order to test intrareader agreement.

## Statistical analyses

Continuous variables are reported as median and interquartile range, and categorical variables as proportions. CT body composition parameters were considered as continuous variables. We calculated Spearman correlation to assess the association among different fat distribution indices as well as between age and CT body composition parameters.

We checked the linearity between continuous predictor variables and the logit of the outcome, and univariate logistic regression analyses were performed to identify the main CT body composition parameters influencing adverse outcomes (hospitalization, MV or death,

death alone) in COVID-19 patients. For these parameters and for each outcome, we applied a multivariate logistic model adjusted for sex, age, and calendar period (in weeks since the beginning of the outbreak). We choose not to adjust for patient conditions at disease onset since these could be causally linked to body composition. Furthermore, we did not adjust for cardiovascular and metabolic pre-existing conditions because they can be mediators in the relationship between body composition and outcomes. Hospitalization, mechanical ventilation (MV) and/or death, and mortality at 40 days odds ratios (OR) with 95% confidence intervals (95% CI) are reported for unit increase of CT body composition parameters (HU for pectoral density and $cm^2$ for adipose tissue variables). Only the OR of IMAT for mortality is reported for IMAT quartiles.

As sensitivity analyses, we restricted the sample to patients with no comorbidities, diabetes only, and cardiovascular comorbidities only.

We also tested the association between body composition and disease severity at ER presentation using the following biomarkers: CRP as an indicator of cytokine storm intensity; SO2 and CT disease extension as indices of the degree of lung parenchyma involvement. The associations were investigated using multivariate linear regression models adjusted for sex, age, and calendar period.

Lastly, we analyzed the relationship between age and body composition parameters. A mediation analysis was conducted to assess to what degree body composition parameters could explain the effect of age on death by using logit model adjusted for sex, age, and calendar period. This analysis subdivided the total effect into indirect effects representing the causal mechanism through body composition, as opposed to direct effects represented by all other mechanisms [37].

Intrareader agreement was evaluated by Spearman's rank correlation coefficient and respective p-value.

Data analysis was performed using Stata 13.0 SE (Stata Corporation, Texas, TX).

## Results

### Study population

Of the 488 RT-PCR-positive patients presenting to the ER in the time period under study, we included 318 consecutive patients (median age 65.7 years, females 37.7%) satisfying the inclusion criteria (Fig 1). The remaining 170 patients did not undergo CT scan primarily because chest X-rays and clinical presentation did not suggest pneumonia, and none of them died or received MV during follow up. Patient characteristics are reported in Table 1. During follow up, 205 (64.47%) hospitalizations and 58 (18.24%) deaths were registered; 68 (21.4%) patients were treated with invasive or non-invasive MV, and a total of 97 (30.5%) patients died or needed MV.

### Body composition parameter selection

**Relationship between CT fat distribution parameters and BMI.** Association of CT fat distribution parameters with BMI was estimated only for patients with an available BMI measured within six months previous to ER presentation (n = 88). Of the CT parameters describing fat distribution, the strongest association with BMI was for TAT (r = 0.706, p<0.001), which we chose over SAT as a measure of general adiposity. TAT was strongly associated with SAT (r = 0.959, p <0.001) while the associations between IMAT and VAT and both BMI and TAT were weaker (S1 Table).

**Distribution of body composition parameters according to outcome.** In a preliminary analysis (S2 Table), we evaluated the association between body composition parameters expressed in quartiles and outcomes, observing a linear relationship of all parameters with

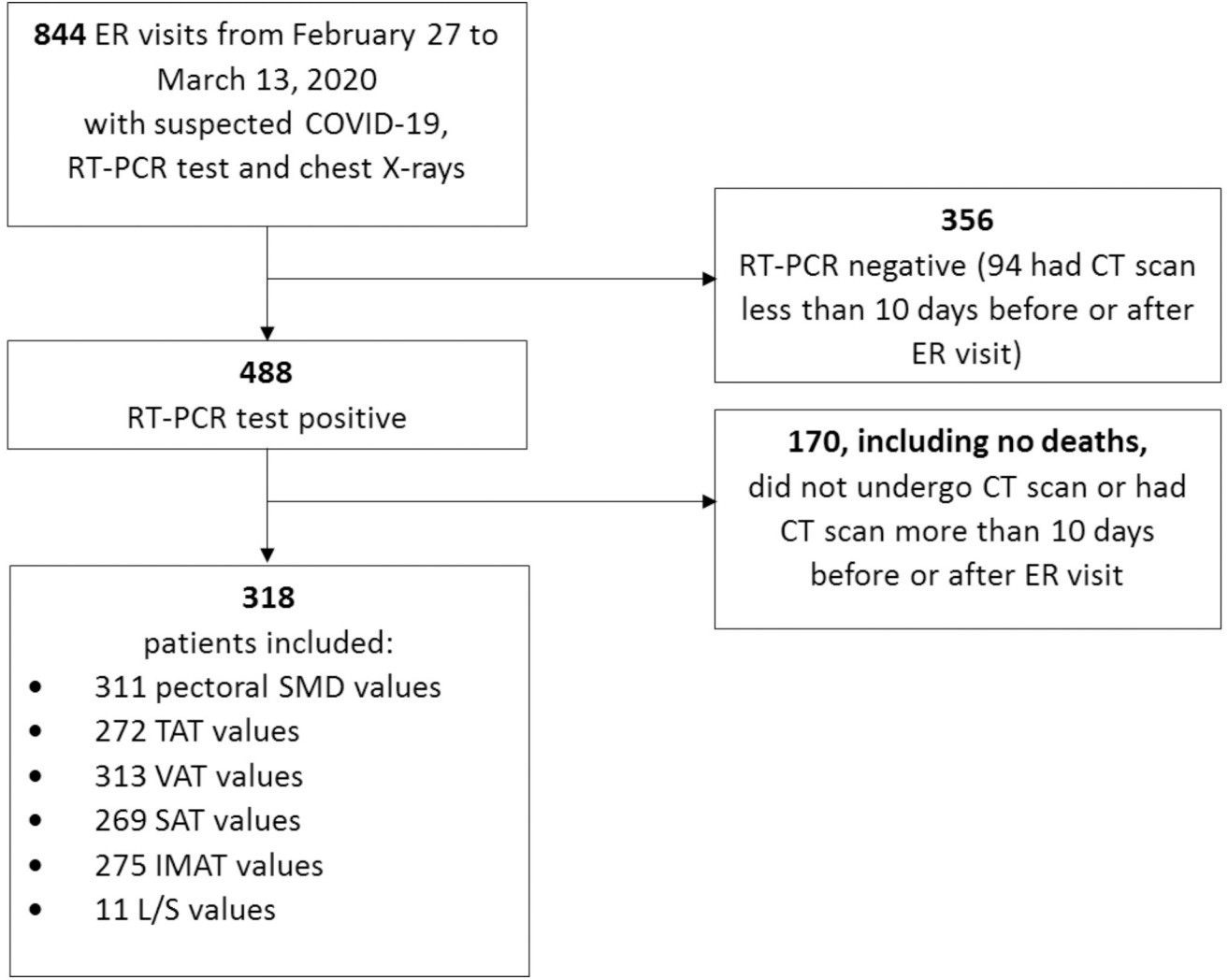

**Fig 1. Flowchart describing patient selection.**

hospitalization and MV or death. For death alone, pectoral muscle density and VAT were linearly associated and almost no association was observed for TAT, while the relationship with IMAT was better described by a model including IMAT quartiles. As no association was found with the three outcomes, pectoral muscle area and liver-to-spleen ratio were dropped in subsequent analyses.

**Intrareader agreement.** Intrareader agreement was excellent for pectoral muscle area and density and for fat compartment areas (Spearman rho between 0.96 and 1.00, p<0.001) and moderate for liver-to-spleen ratio (Spearman rho = 0.78, p = 0.001).

### Associations between body composition parameters and patient outcomes

After correcting for age, sex and calendar period, increased muscle density showed a protective effect on hospitalization (OR for one HU increase = 0.967; 95%CI = 0.935–1.000), death (OR for one HU increase = 0.962; 95%CI = 0.922–1.004) and MV or death (OR for one HU increase = 0.964; 95%CI = 0.934–0.996) (Fig 2). Increased TAT was a risk factor for hospitalization and for MV or death (OR for one $cm^2$ increase = 1.005; 95%CI = 1.002–1.008 and OR

**Table 1. Clinical and body composition parameters in the population as a whole and in patients experiencing different outcomes.**

| Variables | All Patients | Hospitalization | Mechanical Ventilation | Death | Mechanical Ventilation or Death |
|---|---|---|---|---|---|
| | | N (%) | N (%) | N (%) | N (%) |
| | 318 | 205 (64.47) | 68 (21.38) | 58 (18.24) | 97 (30.50) |
| Age (years) | 65.7 (52.8; 75.7) | 71.8 (61.4; 79.8) | 69.8 (63.2; 77.6) | 79.8 (72.5; 85.0) | 73.8 (66.4;82.5) |
| Females | 120 (37.7) | 69 (57.5) | 16 (13.3) | 13 (10.8) | 27 (22.5) |
| Calendar period (Week 1) | 36 (11.3) | 27 (75.0) | 15 (41.7) | 8 (22.2) | 17 (47.2) |
| (Week 2) | 167 (52.5) | 123 (73.7) | 42 (25.2) | 40 (24.0) | 61 (36.5) |
| (Week 3) | 115 (36.16) | 55 (47.8) | 11 (9.6) | 10 (8.7) | 19 (16.5) |
| Charlson Comorbidity Index (0) | 239 (75.16) | 134 (56.1) | 45 (18.8) | 27 (11.3) | 58 (24.3) |
| (1) | 22 (6.92) | 18 (81.8) | 7 (31.8) | 7 (31.8) | 11 (50.0) |
| (2) | 20 (6.29) | 18 (90.0) | 6 (30.0) | 5 (25.0) | 8 (40.0) |
| (3) | 37 (11.64) | 35 (94.6) | 10 (27.0) | 19 (51.4) | 20 (54.1) |
| Diabetes | 43 (13.52) | 41 (95.4) | 20 (46.5) | 11 (25.6) | 23 (53.5) |
| COPD | 10 (3.14) | 10 (100) | 3 (30.0) | 7 (70.0) | 9 (90.0) |
| Dementia | 1 (0.31) | 1 (100) | 1 (100) | 1 (100) | 1 (100) |
| Chronic kidney failure | 3 (0.94) | 3 (100) | 2 (66.7) | 2 (66.7) | 3 (100) |
| Previous cancer diagnosis | 51 (16.04) | 43 (84.3) | 15 (29.4) | 14 (27.5) | 20 (39.2) |
| Hypertension | 56 (17.61) | 49 (87.5) | 21 (37.5) | 20 (35.7) | 27 (48.2) |
| Arrhythmias | 24 (7.55) | 22 (91.7) | 7 (29.2) | 12 (50.0) | 14 (58.3) |
| Cardiovascular diseases | 47 (14.78) | 42 (89.4) | 16 (34.0) | 22 (46.8) | 26 (55.3) |
| Days from symptom onset | 7 (4; 8) | 6 (4;8) | 6 (5; 7) | 5 (2;7) | 5 (3;7) |
| White blood cells (10^9/L) | 5.22 (4.14; 6.63) | 5.59 (4.11; 6.87) | 5.82 (4.17; 7.18) | 6.27 (4.54; 8.05) | 5.86 (4.31; 7.58) |
| Lymphocytes (10^9/L) | 0.96 (0.71; 1.34) | 0.88 (0.68; 1.25) | 0.84 (0.63; 1.00) | 0.78 (0.49; 0.92) | 0.83 (0.61; 1) |
| Neutrophils (10^9/L) | 3.84 (2.95; 4.75) | 4.10 (2.83; 5.29) | 4.57 (2.94; 5.80) | 4.69 (3.50; 6.33) | 4.62 (3.27; 5.82) |
| Platelets (10^9/L) | 176 (142; 219) | 171 (133.27; 219) | 156.5 (129; 190) | 160.1 (124; 201.5) | 159.5 (124; 197.9) |
| C-reactive protein (mg/dL) | 5.34 (2.10; 11.58) | 7.94 (3.60; 13.62) | 11.68 (6.40; 16.00) | 11.35 (4.18; 15.91) | 11.05 (4.79; 15.87) |
| LDH (U/L) | 514.7 (471.0; 594) | 533.8 (482.5; 665.0) | 584.9 (514.6; 742.7) | 534.9 (468.0; 745.2) | 558.0 (499.0; 734.4) |
| SO2 (%) | 94.8 (92.8; 96.1) | 93.7 (91.7; 95.3) | 91.8 (90.0; 94.2) | 92.6 (89.6; 94.5) | 92.4 (90; 94.5) |
| CT extension <20% | 109 (34.28) | 37 (33.9) | 8 (7.3) | 7 (6.4) | 13 (11.9) |
| 20–39% | 115 (36.16) | 82 (71.3) | 21 (18.3) | 14 (12.2) | 30 (26.1) |
| 40–59% | 60 (18.87) | 52 (86.7) | 20 (33.3) | 16 (26.7) | 27 (45.0) |
| ≥60% | 34 (10.69) | 34 (100) | 19 (55.9) | 21 (61.8) | 27 (79.4) |
| Pectoral muscle area (cm$^2$) | 17 (12; 21) | 16 (12; 21) | 15 (12; 20) | 15 (11; 19) | 15 (11; 20) |
| Pectoral muscle density (HU) | 34 (27; 41) | 33 (26; 39) | 32 (22; 40) | 30 (23; 37) | 32.5 (23; 39) |
| L/S ratio | 223.5 (159; 292.5) | 230 (167; 311) | 250.5 (190; 346) | 215.5 (160; 291) | 246.5 (168; 314) |
| TAT (cm$^2$) | 34 (23; 47) | 38 (27; 51) | 46 (33; 57) | 45 (30; 58) | 43.5 (30; 56) |
| VAT (cm$^2$) | 152 (102; 210) | 152 (108.5; 211.5) | 152 (115.5; 220.5) | 122 (99; 179) | 147.5 (112; 210) |
| SAT (cm$^2$) | 27 (18; 37) | 30.5 (21; 42) | 35 (26;45) | 35 (21; 49) | 34 (25; 45) |
| IMAT (cm$^2$) | 223.5 (159; 292.5) | 230 (167; 311) | 250.5 (190; 346) | 215.5 (160; 291) | 246.5 (168; 314) |

Patients' pre-existing conditions, along with clinical, laboratory and chest CT variables at ER presentation, including body composition parameters in the population as a whole, in hospitalized patients, in patients who underwent mechanical ventilation, in those who underwent mechanical ventilation or died, and in those who died. Continuous variables are presented as median (IQR); categorical variables are presented as frequencies (%). Column percentages are reported for all patients and row percentages are reported for subpopulations with each different outcome. Calendar period is expressed in weeks since the beginning of the outbreak. Cardiovascular diseases group heart failure, ischemic cardiopathy, and vascular diseases. COPD, chronic obstructive pulmonary disease; LDH, lactate dehydrogenase; SO2, oxygen saturation level; L/S, liver to spleen; TAT, total adipose tissue area; VAT, visceral adipose tissue area; SAT, subcutaneous adipose tissue area; IMAT, intermuscular adipose tissue area.

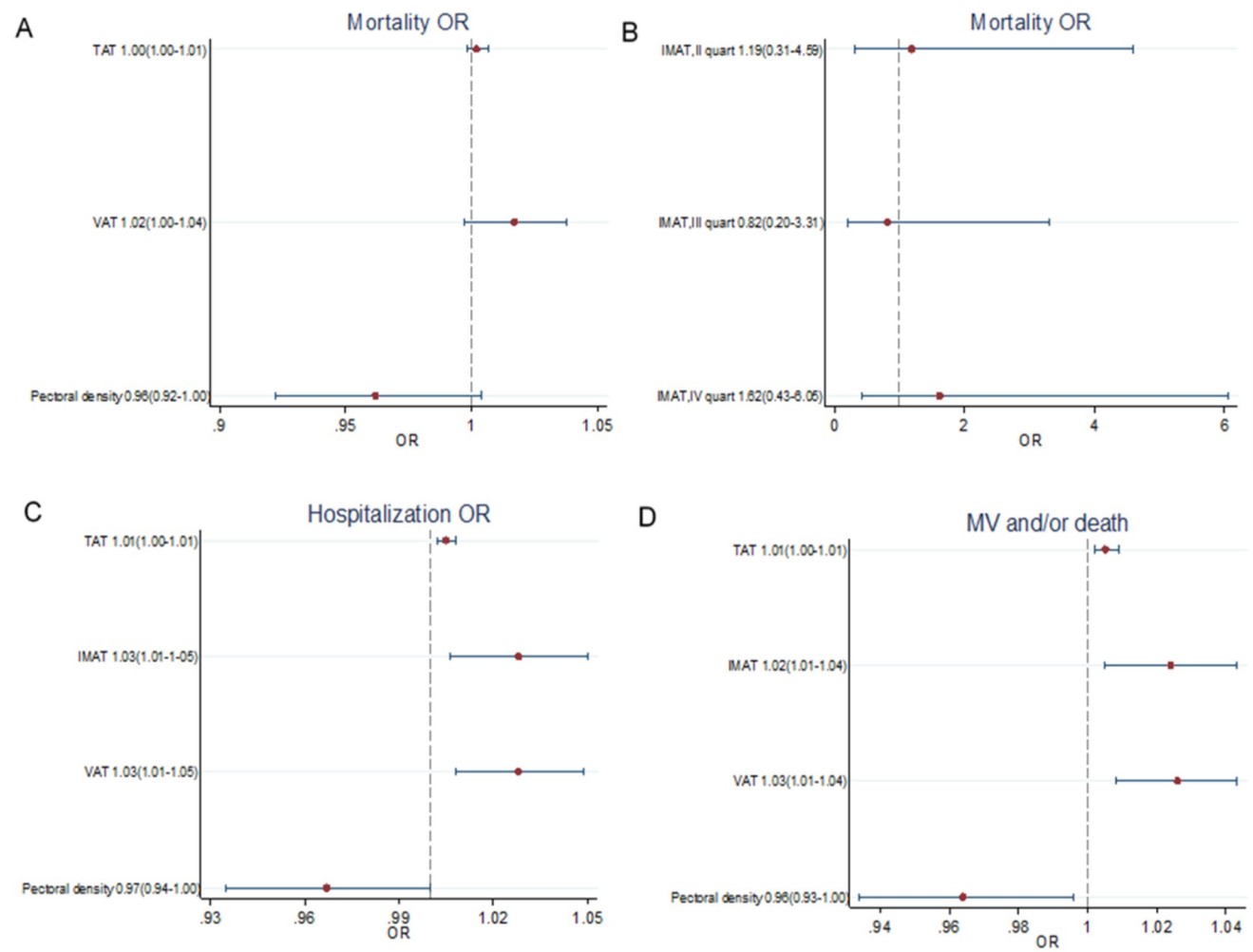

**Fig 2. Multivariate logistic models adjusted for sex, age, and calendar period (weeks since the beginning of the outbreak).** A) Mortality OR for unit increase with 95% CI for unit increase of pectoral muscle density (HU), VAT (cm²), and TAT (cm²). B) Mortality OR with 95% CI for IMAT quartiles (cm²). C) Hospitalization OR with 95% CI for unit increase of pectoral muscle density (HU), VAT (cm²), IMAT (cm²), and TAT (cm²). D) Mechanical ventilation and/or death OR with 95% CI for unit increase of pectoral muscle density (HU), VAT (cm²), IMAT (cm²), and TAT (cm²). OR, Odds Ratio; CI, Confidence Interval; TAT, total adipose tissue area; VAT, visceral adipose tissue area; IMAT, intermuscular adipose tissue area.

for one cm² increase = 1.005; 95%CI = 1.002–1.009, respectively), but only a small excess was appreciable for the risk of death (OR for one cm² increase = 1.002; 95%CI = 0.998–1.007).

Increased VAT and IMAT were significantly associated with hospitalization (OR for one cm² increase = 1.028, 95%CI = 1.008–1.049 and OR for one cm² increase = 1.028, 95%CI = 1.006–1.050, respectively) and MV or death (OR for one cm² increase = 1.026, 95%CI = 1.008–1.043 and OR for one cm² increase = 1.024, 95%CI = 1.005–1.043, respectively). Considering VAT and IMAT as risk factors for death alone, the associations were weaker and the excesses were possibly due to random fluctuations (OR for one cm² increase of VAT = 1.017, 95%CI = 0.997–1.038, OR for the last quartile of IMAT vs. first quartile = 1.615, 95%CI = 0.431–6.053).

Except for VAT, the effect of body composition parameters on outcomes decreased or disappeared when excluding all comorbidities, cardiovascular diseases only, and diabetes only, but not when excluding previous cancer diagnosis only (S3 Table).

## Associations between body composition and disease severity at ER presentation

TAT, VAT, and IMAT in our sample were linearly associated with all secondary outcomes (CRP, SO2, and CT disease extension at ER presentation). Instead, pectoral muscle density was linearly associated only with CT disease extension and SO2 but not with CRP (S2 and S3 Figs). Consequently, in multivariate linear regression models corrected for age, sex and calendar period, all body composition parameters were used as continuous variables, with the exception of pectoral muscle density, which was used in quartiles in the model for CRP.

As reported in Table 2, in multivariable models a decreasing pectoral muscle density was linearly associated with increasing lung involvement (increasing CT disease extension and decreasing SO2), while the second and the fourth pectoral muscle density quartiles were inversely associated with CRP as an indicator of systemic inflammation. Increasing TAT, VAT, and IMAT were associated with increasing CT disease extension and decreasing SO2, while the association with CRP was higher for VAT (R squared 0.12 for VAT and 0.09 for TAT).

## Relationship between CT body composition parameters and age and mediation analysis

As part of the mediation analysis, to determine whether the effect of age on Covid-19 prognosis was at least partially mediated by a worse body composition, we analyzed the association between CT body composition parameters expressed in quartiles and age. Pectoralis muscle density linearly decreased with age. VAT and IMAT increased with age, while the relationship between TAT and age was more complex, without a clear association (S4 Table).

Consequently, a possible mediation effect on death was evaluated for VAT, IMAT, and pectoralis muscle density, after correcting for sex and calendar period. No mediation effect was found for VAT and IMAT, even if they were associated with both age and death. Instead, the effect of age on death decreased when adding pectoralis muscle density to the model. This analysis suggests that approximately 3% of the effect of age on death was mediated by decreased muscle density (Fig 3).

**Table 2. Association of body composition parameters with biomarkers of disease progression at ER presentation.**

| Variables | CRP | | SO2 | | CT disease extension | |
|---|---|---|---|---|---|---|
| | β | 95% CI | β | 95% CI | β | 95% CI |
| Pectoral density (quart1: 3–27] | 0 | | | | | |
| (quart2: 28–34] | -3.648 | -5.760; -1.535 | | | | |
| (quart3: 35–41] | -2.518 | -4.733; -.304 | | | | |
| (quart4: 41.1–63] | -4.820 | -7.238; -2.403 | | | | |
| Pectoral density[a] | | | 0.058 | 0.000; 0.116 | -0.485 | -0.719; -0.252 |
| TAT[a] | 0.008 | 0.000; 0.016 | -0.006 | -0.011; -0.001 | 0.046 | 0.025; 0.068 |
| VAT[a] | 0.064 | 0.017; 0.111 | -0.033 | -0.064; -0.003 | 0.258 | 0.136; 0.381 |
| IMAT[a] | 0.038 | -0.016; 0.092 | -0.036 | -0.071; 0.000 | 0.245 | 0.106; 0.384 |

Multivariate linear regression models adjusted for sex, age and calendar period depicting the associations between pectoral density, TAT, VAT, and IMAT with disease severity at ER presentation described by CRP, SO2, and CT extension. TAT, total adipose tissue area; VAT, visceral adipose tissue area; IMAT, intermuscular adipose tissue area tissue area; ER, Emergency room; CRP, C-reactive protein; SO2, oxygen saturation level.

[a]for unit increase

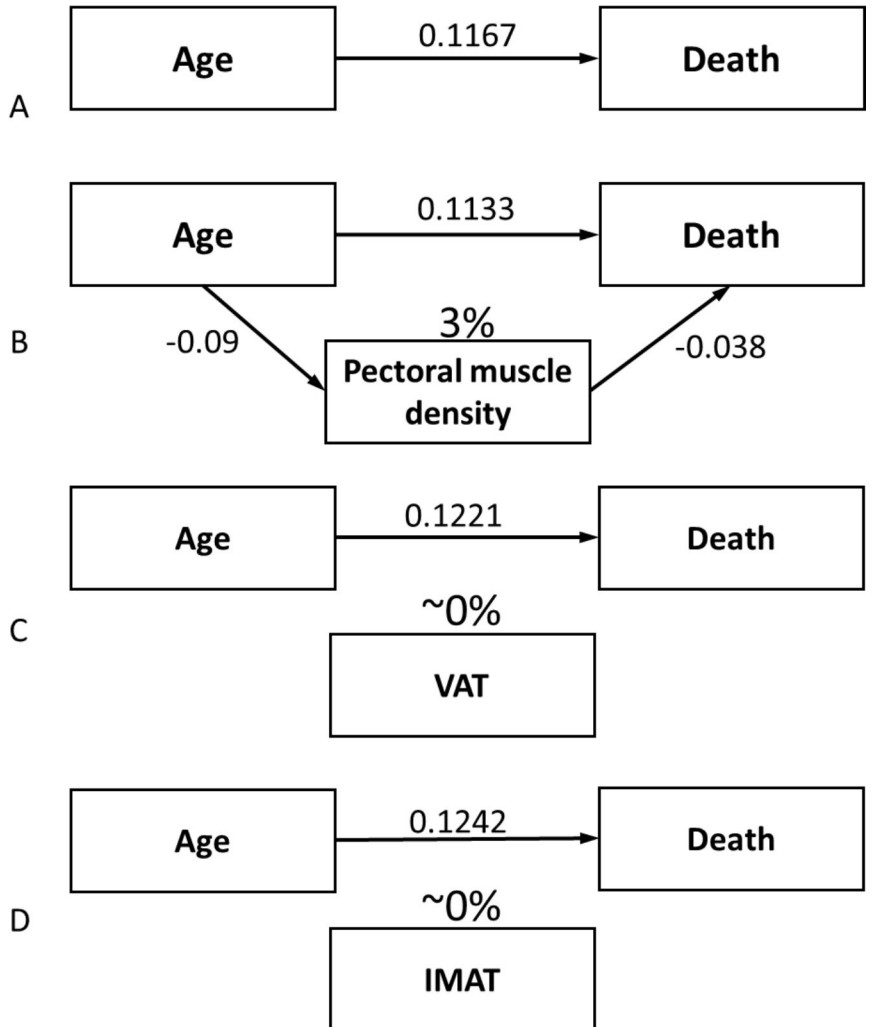

**Fig 3. Mediation analysis.** A) β coefficient of the relationship between age and the logit of death, after correcting for sex and calendar period. B) The coefficient decreases when adding pectoral muscle density to the model, indicating that about 3% of the effect of age on death is mediated by pectoral muscle quality. Vice versa, the coefficient does not decrease when adding VAT (C) or IMAT (D) to the model, suggesting that a mediation effect does not exist for ectopic fat on the relationship between age and death. VAT, visceral adipose tissue area; IMAT, intermuscular adipose tissue area.

## Discussion

This observational study showed an association of chest CT measures of fat distribution and muscle quality with a continuum of outcomes representing COVID-19 progression. Chest CT scans routinely performed in symptomatic COVID-19 patients at ER presentation were used to generate body composition measures. Increasing thoracic TAT as a measure of general adiposity as well as VAT and IMAT, representing ectopic fat compartments, were associated with increased risk of hospitalization, the composite of death and MV, and, to a lesser degree, death alone. A higher pectoral density, representing better muscle quality, exhibited a protective effect on the same outcomes. Pectoral muscle area, and liver-to-spleen ratio as a measure of liver steatosis, were not associated with these outcomes.

A few studies have validated thoracic CT to assess ectopic fat areas [38], and pectoral muscle area and density have been used to study the effect of muscle wasting on the outcomes of different diseases [30–32].

Our data are consistent with recently published studies on COVID-19 patients. In a small observational study of 51 patients, a predictive model for hospitalization including VAT and SAT measured on abdominal CT along with clinical variables, performed better than the model that included clinical variables only [39], while, in a study of 165 patients, increasing abdominal VAT was associated with MV or death [23]. In another small cohort of hospitalized patients, increasing upper abdominal VAT on chest CT was associated with higher risk of intensive care admission or MV [19], while in a study of 150 patients who performed chest CT at the ER, upper abdominal VAT was independently associated with the need for intensive care [20]. Higher VAT and lower lumbar skeletal muscle density on abdominal CT of 143 hospitalized COVID-19 patients were independently associated with critical illness [5]. Finally, in a small study of 58 patients, an increasing ratio between waist circumference (as a measure of fat) and paraspinal muscle circumference (as a measure of muscle), measured at T12 level, was associated with a higher probability of MV [40]. In comparison with the present investigation, all cited studies were conducted on smaller cohorts, for the most part including hospitalized patients only, and with a restricted spectrum of intermediate and final outcomes. Furthermore, in some of these studies body composition parameters were measured on patients undergoing abdominal CT scans for specific indications (e.g., abdominal pain), thus in a selected population with specific clinical characteristics [5, 39]. Besides the impact of body composition parameters on COVID-19 progression, we investigated their association with biomarkers of disease severity at ER presentation, trying to distinguish between the two main courses of COVID-19 progression: lung involvement and inflammatory response. Decreasing pectoral muscle density and increasing TAT, VAT, and IMAT were all associated with lung parenchyma involvement reflected by SO2 and CT extension, while higher VAT showed the strongest association with CRP, a proxy of the systemic inflammatory response.

While the impact of other CT body composition parameters on COVID-19 outcomes was much less evident when excluding patients with comorbidities, the effect of VAT remained substantially similar. These sensitivity analyses suggest that body composition and comorbidities, which are linked in a complex interplay, may be on the same causal sequence in determining COVID-19 outcomes, with the exception of VAT, which may also act through different pathways.

Overall, our results confirm the association between adipose tissue, especially ectopic fat, and the inflammatory state driving disease severity and progression in COVID-19. VAT is known to be an endocrine organ with pro-inflammatory characteristics [13, 21] and different studies have measured higher levels of circulating inflammatory cytokines in people with visceral adiposity compared with lean individuals [13], leading to the hypothesis that they are susceptible to developing a more powerful cytokine storm during COVID-19 progression [41]. Moreover, abdominal obesity can profoundly alter pulmonary function by diminishing exercise capacity and augmenting airway resistance, resulting in increased respiratory fatigue [42]. Also, pectoral muscle density is a measure of respiratory muscle capacity, which is of central importance in COVID-19 patients undergoing MV. In fact, in these patients, death is frequently the consequence of muscle fatigue.

The risk of muscular insufficiency increases with age, since sarcopenia is one of the main hallmarks of ageing [15, 43]. Accordingly, pectoral muscle density in our study decreased with age, while VAT and IMAT increased. Even if associated with muscle deterioration and function [30, 31], IMAT is still a measure of ectopic fat deposition rather than a measure of the quality of the muscle itself. This association with age, along with the association between these parameters and COVID-19 outcomes, justified our choice to explore the possibility of a mediating effect of body composition on the strong relationship between age and COVID-19 outcome. We found that approximately 3% of the effect of age on death was mediated by

decreased pectoral muscle density, while no mediation effect was found for VAT or IMAT, leading to the hypothesis that ectopic fat may belong to another causal pathway than that linking age, muscle quality, and death. This opens up the way for new study hypotheses on the pathogenetic mechanisms in COVID-19 disease progression.

This study has several limitations. First, data collection was retrospective and chest CT body composition parameters are less validated than those obtained from abdominal CT at L3 level [18, 27, 28]. Body composition was collected at ER admission, i.e., 2 to 10 days after symptom onset. Therefore, we cannot exclude that body composition was already altered by disease progression, leading to an inversion of the cause-effect interpretation. In fact, patients with more severe forms of COVID-19 may experience loss of muscle mass [44, 45]. Nevertheless, the median time from symptom onset and ER visit in our study was 7 days, a very short time to see important changes in CT-measured body composition.

As data on patient height were mostly lacking, it was not possible to calculate the skeletal muscle index, a marker of muscle quantity more reliable than the skeletal muscle area. For this reason, we may argue that pectoral muscle quality seems to be more meaningful than pectoral muscle quantity. However, more studies are needed, especially because a recent study showed that lower pectoral muscle area and index were associated with COVID-19 outcomes [22]. Due to the lack of data on height, BMI was available only in a subset of patients. However, TAT allowed us to have a reliable measure of general adiposity, and ectopic fat depots, particularly VAT, are generally stronger outcome predictors than BMI when studying cardiometabolic risk and associated systemic inflammatory state [46].

We had to exclude 170 COVID-19 patients for whom CT was not available, mostly because chest X-rays and clinical presentation did not suggest pneumonia. Consequently, some potentially eligible patients were missed. This may have occurred among the less severe cases, who were referred to household isolation. This may have introduced a selection bias, especially if we envisage a possible role of body composition and above all obesity as a known risk factor for severe disease, in the decision to perform CT scans. This bias may have led to an underestimation of the association between body composition and patient outcomes.

Finally, the described associations may not be strong enough to be used as prognostic biomarkers in guiding clinical decision making.

## Conclusion

In conclusion, we confirmed the association between low muscle quality and ectopic fat accumulation with COVID-19 severity and outcomes. VAT was particularly associated with inflammatory reaction in COVID-19, while all indices including pectoral muscle density were associated with parenchymal involvement. Low muscle quality appears to be one of the mechanisms for the extremely strong effect of age on COVID-19 mortality.

Despite the limits of this observational study, the consistency of results observed on different outcomes and indicators, including disease severity markers and medium-term outcomes, together with the results of previous smaller studies, make a causal relation plausible.

## Supporting information

**S1 Table. Correlations between BMI and CT fat distribution parameters.** Spearman correlations between BMI and CT fat distribution parameters. Spearman's rank correlation coefficients are reported with respective p-values between brackets. CT: Computed Tomography; IMAT: intermuscular adipose tissue area; SAT: subcutaneous adipose tissue area; TAT: total adipose tissue area; VAT: visceral adipose tissue area.
(PDF)

**S2 Table. Preliminary analysis to verify the relationship and linearity between CT body composition parameters (quartiles) and the outcome, by an unadjusted logit regression model.** β coefficients with respective 95% confidence intervals are reported. CT: Computed Tomography; IMAT: intermuscular adipose tissue area; L/S: liver to spleen ratio; TAT: total adipose tissue area; VAT: visceral adipose tissue area.
(PDF)

**S3 Table. Multivariate logistic model adjusted for sex, age, and calendar period (weeks since the beginning of the outbreak).** Hospitalization, ventilation and/or death, and mortality OR with 95% CI are reported for unit increase of CT body composition parameters (HU for pectoral density and cm$^2$ for VAT and TAT) and for IMAT quartiles. The same model is reported after excluding all patients with comorbidities, only patients with diabetes, only patients with cardiovascular comorbidities, including hypertension, and only patients with previous cancer diagnosis. OR adj: adjusted for age, sex, and calendar period. HU: Hounsfield Unit; IMAT: intermuscular adipose tissue area; TAT: total adipose tissue area; VAT: visceral adipose tissue area. [a] as continuous variable (for one unit increase).
(PDF)

**S4 Table. Distribution of CT body composition parameters expressed in quartiles in different age quartiles.** P[*] Pearson's chi-squared test and p-value for the hypothesis of independence in the two-way table. IMAT: intermuscular adipose tissue area; TAT: total adipose tissue area; VAT: visceral adipose tissue area.
(PDF)

**S1 Fig. Representative images for different CT body composition parameters.** Total adipose tissue, TAT (A), subcutaneous adipose tissue, SAT (B), visceral adipose tissue, VAT (C), intermuscular adipose tissue, IMAT (D), were all measured at the level of T7-T8 vertebrae. Pectoral muscle area and density were measured on the right side at a level immediately superior to the aortic arch (E).
(PDF)

**S2 Fig. Regression line (red) and locally weighted scatterplot smoothing (lowess) smoother curve (green) superimposed on the scatter diagram for pectoral density and CRP, CT disease extension, SO2, and for TAT and CRP, CT disease extension, SO2.** CRP: C-reactive protein; CT: Computed Tomography; SO2: oxygen saturation level; TAT: total adipose tissue area.
(PDF)

**S3 Fig. Regression line (red) and lowess smoother curve (green) superimposed on the scatter diagram for VAT and CRP, CT disease extension, SO2, and for IMAT and CRP, CT disease extension, SO2.** CRP: C-reactive protein; CT: Computed Tomography; IMAT: intermuscular adipose tissue area SO2: oxygen saturation level; VAT: visceral adipose tissue area.
(PDF)

## Acknowledgments

We thank Jacqueline Costa for the English language editing. The following are the members of the Reggio Emilia COVID-19 Working Group: Massimo Costantini, Roberto Grilli, Massimiliano Marino, Giulio Formoso, Debora Formisano, Paolo Giorgi Rossi, Emanuela Bedeschi, Cinzia Perilli, Elisabetta La Rosa, Eufemia Bisaccia, Ivano Venturi, Massimo Vicentini, Cinzia

Campari, Francesco Gioia, Serena Broccoli, Marta Ottone, Pierpaolo Pattacini, Giulia Besutti, Valentina Iotti, Lucia Spaggiari, Pamela Mancuso, Andrea Nitrosi, Marco Foracchia, Rossana Colla, Alessandro Zerbini, Marco Massari, Anna Maria Ferrari, Mirco Pinotti, Nicola Facciolongo, Ivana Lattuada, Laura Trabucco, Stefano De Pietri, Giorgio Francesco Danelli, Laura Albertazzi, Enrica Bellesia, Simone Canovi, Mattia Corradini, Tommaso Fasano, Elena Magnani, Annalisa Pilia, Alessandra Polese, Silvia Storchi Incerti, Piera Zaldini, Efrem Bonelli, Bonanno Orsola, Matteo Revelli, Carlo Salvarani, Carmine Pinto, Francesco Venturelli, Elisabetta Teopompi, Annalisa Gallina, Annalisa Bertellini, Stefania Costi, Stefania Fugazzaro.

## Author Contributions

**Conceptualization:** Giulia Besutti, Massimo Pellegrini, Pierpaolo Pattacini, Carlo Salvarani, Giovanni Guaraldi, Paolo Giorgi Rossi.

**Data curation:** Giulia Besutti, Marta Ottone, Michele Cantini, Jovana Milic, Efrem Bonelli, Giovanni Dolci, Giulia Cassone, Guido Ligabue, Lucia Spaggiari, Tommaso Fasano, Simone Canovi, Marco Massari, Giovanni Guaraldi.

**Formal analysis:** Marta Ottone, Efrem Bonelli.

**Investigation:** Massimo Pellegrini, Marta Ottone, Michele Cantini, Jovana Milic, Efrem Bonelli, Giovanni Dolci, Giulia Cassone, Lucia Spaggiari, Simone Canovi.

**Methodology:** Paolo Giorgi Rossi.

**Project administration:** Giulia Besutti, Massimo Pellegrini.

**Supervision:** Massimo Pellegrini.

**Writing – original draft:** Giulia Besutti, Massimo Pellegrini, Marta Ottone, Giovanni Guaraldi, Paolo Giorgi Rossi.

**Writing – review & editing:** Giulia Besutti, Massimo Pellegrini, Guido Ligabue, Pierpaolo Pattacini, Tommaso Fasano, Marco Massari, Carlo Salvarani, Giovanni Guaraldi, Paolo Giorgi Rossi.

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
