## [Decision Letter · Decision Letter 0]

13 Apr 2021

PONE-D-21-05721

The impact of chest CT body composition parameters on clinical outcomes in COVID-19 patients

PLOS ONE

Dear Dr. Massimo,

Thank you for submitting your manuscript to PLOS ONE. After careful consideration, we feel that it has merit but does not fully meet PLOS ONE’s publication criteria as it currently stands. Therefore, we invite you to submit a revised version of the manuscript that addresses the points raised during the review process.

We look forward to receiving your revised manuscript.

Kind regards,

Francesco Di Gennaro

Academic Editor

PLOS ONE

Journal Requirements:

Please provide additional details regarding participant consent. In the ethics statement in the Methods and online submission information, please ensure that you have specified what type you obtained (for instance, written or verbal, and if verbal, how it was documented and witnessed). If your study included minors, state whether you obtained consent from parents or guardians. If the need for consent was waived by the ethics committee, please include this information.

We note that you have indicated that data from this study are available upon request. PLOS only allows data to be available upon request if there are legal or ethical restrictions on sharing data publicly. For information on unacceptable data access restrictions, please see http://journals.plos.org/plosone/s/data-availability#loc-unacceptable-data-access-restrictions.

3a) If there are ethical or legal restrictions on sharing a de-identified data set, please explain them in detail (e.g., data contain potentially identifying or sensitive patient information) and who has imposed them (e.g., an ethics committee). Please also provide contact information for a data access committee, ethics committee, or other institutional body to which data requests may be sent.

3b) If there are no restrictions, please upload the minimal anonymized data set necessary to replicate your study findings as either Supporting Information files or to a stable, public repository and provide us with the relevant URLs, DOIs, or accession numbers. Please see http://www.bmj.com/content/340/bmj.c181.long for guidelines on how to de-identify and prepare clinical data for publication. For a list of acceptable repositories, please see http://journals.plos.org/plosone/s/data-availability#loc-recommended-repositories.

Thank you for stating the following in the Acknowledgments Section of your manuscript:

We thank Jacqueline Costa for the English language editing. This study is part of a larger

project supported by Ministry of Health (Grant number COVID-2020-12371808).

The following are the members of the Reggio Emilia COVID-19 Working Group: Massimo

Costantini, Roberto Grilli, Massimiliano Marino, Giulio Formoso, Debora Formisano, Paolo

Giorgi Rossi, Emanuela Bedeschi, Cinzia Perilli, Elisabetta La Rosa, Eufemia Bisaccia, Ivano

Venturi, Massimo Vicentini, Cinzia Campari, Francesco Gioia, Serena Broccoli, Marta Ottone,

Pierpaolo Pattacini, Giulia Besutti, Valentina Iotti, Lucia Spaggiari, Pamela Mancuso, Andrea

Nitrosi, Marco Foracchia, Rossana Colla, Alessandro Zerbini, Marco Massari, Anna Maria

Ferrari, Mirco Pinotti, Nicola Facciolongo, Ivana Lattuada, Laura Trabucco, Stefano De Pietri,

Giorgio Francesco Danelli, Laura Albertazzi, Enrica Bellesia, Simone Canovi, Mattia

Corradini, Tommaso Fasano, Elena Magnani, Annalisa Pilia, Alessandra Polese, Silvia Storchi

Incerti, Piera Zaldini, Efrem Bonelli, Bonanno Orsola, Matteo Revelli, Carlo Salvarani,

Carmine Pinto, Francesco Venturelli, Elisabetta Teopompi, Annalisa Gallina, Annalisa

Bertellini, Stefania Costi, Stefania Fugazzaro.

The authors received no specific funding for this work.

Additional Editor Comments:

dear authors follow reviewer suggestions to improve your paper

Reviewers' comments:

Reviewer's Responses to Questions

**Comments to the Author**

1. Is the manuscript technically sound, and do the data support the conclusions?

Reviewer #1: Yes

Reviewer #2: Yes

Reviewer #3: Yes

2. Has the statistical analysis been performed appropriately and rigorously? 

Reviewer #1: Yes

Reviewer #2: Yes

Reviewer #3: Yes

3. Have the authors made all data underlying the findings in their manuscript fully available?

Reviewer #1: Yes

Reviewer #2: Yes

Reviewer #3: Yes

4. Is the manuscript presented in an intelligible fashion and written in standard English?

Reviewer #1: Yes

Reviewer #2: Yes

Reviewer #3: Yes

5. Review Comments to the Author

Reviewer #1: The manuscript was good written and discussed.

It can be accepted.

Introduction: was good written

Materials and Methods: were appropriately written and explained briefly

Results: were good written

Discussion: was good written

Reviewer #2: Congratulations

Reviewer #3: This is a relevant study and performed in a correct way scientifically. It addresses relevant research questions and analyses these in an appropriate way. The weaknesses are that it is retrospective and interpolates abdominal body composition parameters from CT slices of the chest. All data are available in the article and the supplementary tables and figures. There are some information lacking in the legends, e.g. what information is given within the brackets in Table S1 and what test i applied in Table S2?

6. PLOS authors have the option to publish the peer review history of their article (what does this mean?). If published, this will include your full peer review and any attached files.

Reviewer #1: **Yes: **Abd El Raouf, Mustafa

Reviewer #2: No

Reviewer #3: No

---

## [Author Response · Author response to Decision Letter 0]

16 Apr 2021

PONE-D-21-05721

The impact of chest CT body composition parameters on clinical outcomes in COVID-19 patients

PLOS ONE

Dear Dr. Massimo,

Thank you for submitting your manuscript to PLOS ONE. After careful consideration, we feel that it has merit but does not fully meet PLOS ONE’s publication criteria as it currently stands. Therefore, we invite you to submit a revised version of the manuscript that addresses the points raised during the review process.

We look forward to receiving your revised manuscript.

Kind regards,

Francesco Di Gennaro

Academic Editor

PLOS ONE

Journal Requirements:

RE: these requirements have been checked.

RE: the consent was verbal, when collected. Given the retrospective nature of the study, it was not always collected. A sentence has been added to better explain these points. 

“Given the retrospective nature of the study, the Ethics Committee authorizes the use of a patient’s data without his/ her written informed consent if all reasonable efforts have been made to contact that patient.”

If needed, we can upload a copy of the written consent and a copy of the declaration which substitutes the informed consent in specific cases (death or impossibility to contact the patient). 

3a) If there are ethical or legal restrictions on sharing a de-identified data set, please explain them in detail (e.g., data contain potentially identifying or sensitive patient information) and who has imposed them (e.g., an ethics committee). Please also provide contact information for a data access committee, ethics committee, or other institutional body to which data requests may be sent.

3b) If there are no restrictions, please upload the minimal anonymized data set necessary to replicate your study findings as either Supporting Information files or to a stable, public repository and provide us with the relevant URLs, DOIs, or accession numbers. Please see http://www.bmj.com/content/340/bmj.c181.long for guidelines on how to de-identify and prepare clinical data for publication. For a list of acceptable repositories, please see http://journals.plos.org/plosone/s/data-availability#loc-recommended-repositories.

RE: According to Italian law, anonymized data can only be made publicly available if there is potential for the re-identification of individuals (https://www.garanteprivacy.it). Furthermore, property of the data remains of the patient, who gave consent to use data for the objective of the study. Thus, the data underlying this study are available on request to researchers who intend to conduct research in the respect of confidentiality (even if anonymous data are provided, they should be published in aggregated form) and for studies with objectives consistent with those of the original study. In order to obtain data, approval must be obtained from the Area Vasta Emilia Nord (AVEN) Ethics Committee, who would check the consistency of the objective and planned analyses and would then authorize us to provide aggregated or anonymized data. Data access requests should be addressed to the Ethics Committee at CEReggioemilia@ausl.re.it as well as to the authors at the Epidemiology unit of AUSL - IRCCS of Reggio Emilia at info.epi@ausl.re.it, who are the data guardians.

The lead author (PGR) affirms that the manuscript is an honest, accurate, and transparent account of the study being reported; that no important aspect of the study has been omitted; and that any discrepancies from the study as planned have been explained.

Thank you for stating the following in the Acknowledgments Section of your manuscript:

We thank Jacqueline Costa for the English language editing. This study is part of a larger

project supported by Ministry of Health (Grant number COVID-2020-12371808).

The following are the members of the Reggio Emilia COVID-19 Working Group: Massimo

Costantini, Roberto Grilli, Massimiliano Marino, Giulio Formoso, Debora Formisano, Paolo

Giorgi Rossi, Emanuela Bedeschi, Cinzia Perilli, Elisabetta La Rosa, Eufemia Bisaccia, Ivano

Venturi, Massimo Vicentini, Cinzia Campari, Francesco Gioia, Serena Broccoli, Marta Ottone,

Pierpaolo Pattacini, Giulia Besutti, Valentina Iotti, Lucia Spaggiari, Pamela Mancuso, Andrea

Nitrosi, Marco Foracchia, Rossana Colla, Alessandro Zerbini, Marco Massari, Anna Maria

Ferrari, Mirco Pinotti, Nicola Facciolongo, Ivana Lattuada, Laura Trabucco, Stefano De Pietri,

Giorgio Francesco Danelli, Laura Albertazzi, Enrica Bellesia, Simone Canovi, Mattia

Corradini, Tommaso Fasano, Elena Magnani, Annalisa Pilia, Alessandra Polese, Silvia Storchi

Incerti, Piera Zaldini, Efrem Bonelli, Bonanno Orsola, Matteo Revelli, Carlo Salvarani,

Carmine Pinto, Francesco Venturelli, Elisabetta Teopompi, Annalisa Gallina, Annalisa

Bertellini, Stefania Costi, Stefania Fugazzaro.

The authors received no specific funding for this work.

RE: Data collection started in 2020 without any funding source. However, final analyses (and future publication) have been partially funded by the Italian Ministry of Health, Grant number COVID-2020-12371808. For this reason, we wrote “This study is part of a larger project supported by Ministry of Health (Grant number COVID-2020-12371808).” in the acknowledgment section. In the new version we have removed this sentence from the acknowledgment section but we should probably change the Funding Statement explaining this partial funding. 

RE: we have added an erratum in ref number 2.

Additional Editor Comments:

dear authors follow reviewer suggestions to improve your paper

Reviewers' comments:

Reviewer's Responses to Questions

Comments to the Author

1. Is the manuscript technically sound, and do the data support the conclusions?

Reviewer #1: Yes

Reviewer #2: Yes

Reviewer #3: Yes

2. Has the statistical analysis been performed appropriately and rigorously?

Reviewer #1: Yes

Reviewer #2: Yes

Reviewer #3: Yes

3. Have the authors made all data underlying the findings in their manuscript fully available?

Reviewer #1: Yes

Reviewer #2: Yes

Reviewer #3: Yes

4. Is the manuscript presented in an intelligible fashion and written in standard English?

Reviewer #1: Yes

Reviewer #2: Yes

Reviewer #3: Yes

5. Review Comments to the Author

Reviewer #1: The manuscript was good written and discussed.

It can be accepted.

Introduction: was good written

Materials and Methods: were appropriately written and explained briefly

Results: were good written

Discussion: was good written

Reviewer #2: Congratulations

Reviewer #3: This is a relevant study and performed in a correct way scientifically. It addresses relevant research questions and analyses these in an appropriate way. The weaknesses are that it is retrospective and interpolates abdominal body composition parameters from CT slices of the chest. All data are available in the article and the supplementary tables and figures. There are some information lacking in the legends, e.g. what information is given within the brackets in Table S1 and what test i applied in Table S2?

RE: We thank the Reviewers for the overall positive judgement of our work. We also thank the Reviewers for the suggestion to improve the manuscript. 

In the new version we have added a sentence to the limitation section: “First, data collection was retrospective and chest CT body composition parameters are less validated than those obtained from abdominal CT at L3 level [18, 27-28].”

We have also added the lacking information to the legends in S1 “Spearman's rank correlation coefficients are reported with respective p-values between brackets.” and S2 “… by an unadjusted logit regression model. β coefficients with respective 95% confidence intervals are reported.”

6. PLOS authors have the option to publish the peer review history of their article (what does this mean?). If published, this will include your full peer review and any attached files.

Do you want your identity to be public for this peer review? For information about this choice, including consent withdrawal, please see our Privacy Policy.

Reviewer #1: Yes: Abd El Raouf, Mustafa

Reviewer #2: No

Reviewer #3: No

---

## [Decision Letter · Decision Letter 1]

3 May 2021

The impact of chest CT body composition parameters on clinical outcomes in COVID-19 patients

PONE-D-21-05721R1

Dear Dr. Pellegrini,

We’re pleased to inform you that your manuscript has been judged scientifically suitable for publication and will be formally accepted for publication once it meets all outstanding technical requirements.

Kind regards,

Francesco Di Gennaro

Academic Editor

PLOS ONE

Additional Editor Comments (optional):

congratulations

Reviewers' comments:

Reviewer's Responses to Questions

**Comments to the Author**

1. If the authors have adequately addressed your comments raised in a previous round of review and you feel that this manuscript is now acceptable for publication, you may indicate that here to bypass the “Comments to the Author” section, enter your conflict of interest statement in the “Confidential to Editor” section, and submit your "Accept" recommendation.

Reviewer #1: All comments have been addressed

Reviewer #3: (No Response)

2. Is the manuscript technically sound, and do the data support the conclusions?

Reviewer #1: Yes

Reviewer #3: (No Response)

3. Has the statistical analysis been performed appropriately and rigorously? 

Reviewer #1: Yes

Reviewer #3: (No Response)

4. Have the authors made all data underlying the findings in their manuscript fully available?

Reviewer #1: Yes

Reviewer #3: (No Response)

5. Is the manuscript presented in an intelligible fashion and written in standard English?

Reviewer #1: Yes

Reviewer #3: (No Response)

6. Review Comments to the Author

Reviewer #1: Thanks for this good study

Introduction was good written

Materials and Methods were good written

Results were good written and qualified

Discussion was good written

Conclusion was clear

Reviewer #3: (No Response)

7. PLOS authors have the option to publish the peer review history of their article (what does this mean?). If published, this will include your full peer review and any attached files.

Reviewer #1: **Yes: **Mustafa Abd El Raouf

Reviewer #3: No

---

## [Editor Report · Acceptance letter]

6 May 2021

PONE-D-21-05721R1 

The impact of chest CT body composition parameters on clinical outcomes in COVID-19 patients 

Dear Dr. Pellegrini:

I'm pleased to inform you that your manuscript has been deemed suitable for publication in PLOS ONE. Congratulations! Your manuscript is now with our production department. 

Kind regards, 

on behalf of

Dr. Francesco Di Gennaro 

Academic Editor

PLOS ONE